# Applications of Titanium Dioxide Nanostructure in Stomatology

**DOI:** 10.3390/molecules27123881

**Published:** 2022-06-17

**Authors:** Shuang Liu, Xingzhu Chen, Mingyue Yu, Jianing Li, Jinyao Liu, Zunxuan Xie, Fengxiang Gao, Yuyan Liu

**Affiliations:** 1Department of Endodontics, Hospital of Stomatology, Jilin University, Changchun 130000, China; liushuang20@jlu.edu.cn (S.L.); chenxz19@jlu.edu.cn (X.C.); yumy20@jlu.edu.cn (M.Y.); jnli20@jlu.edu.cn (J.L.); jinyao21@jlu.edu.cn (J.L.); xiezx21@jlu.edu.cn (Z.X.); 2Changchun Institute of Applied Chemistry, Chinese Academy of Sciences, Changchun 130000, China

**Keywords:** titanium dioxide, nanostructure, dental implant surface modification, antibacterial, dental material additives

## Abstract

Breakthroughs in the field of nanotechnology, especially in nanochemistry and nanofabrication technologies, have been attracting much attention, and various nanomaterials have recently been developed for biomedical applications. Among these nanomaterials, nanoscale titanium dioxide (nano-TiO_2_) has been widely valued in stomatology due to the fact of its excellent biocompatibility, antibacterial activity, and photocatalytic activity as well as its potential use for applications such as dental implant surface modification, tissue engineering and regenerative medicine, drug delivery carrier, dental material additives, and oral tumor diagnosis and treatment. However, the biosafety of nano-TiO_2_ is controversial and has become a key constraint in the development of nano-TiO_2_ applications in stomatology. Therefore, in this review, we summarize recent research regarding the applications of nano-TiO_2_ in stomatology, with an emphasis on its performance characteristics in different fields, and evaluations of the biological security of nano-TiO_2_ applications. In addition, we discuss the challenges, prospects, and future research directions regarding applications of nano-TiO_2_ in stomatology that are significant and worthy of further exploration.

## 1. Introduction

The oral cavity is susceptible to a variety of biological, physical, chemical, and mechanical stimulations due to the fact of its dynamic and open characteristics. The hard and soft tissues of the oral cavity create an ideal environment for microbial growth and biofilm formation, making it prone to various oral diseases such as tooth and dental pulp diseases, tooth loss, periodontal disease, oral mucosal disease, tumor, and trauma. Therefore, there is a need to find a type of material that can meet the requirements for treating various oral diseases.

Recently, biomedical applications of nanomaterials have received considerable attention from researchers. Nanoscale titanium dioxide (nano-TiO_2_) has been widely used in environmental protection, cosmetics, antibacterial agents, and composite nanofillers [1,2]; due to the fact of its unique size and high specific surface area, nano-TiO_2_ has more stable physical and chemical properties compared to titanium dioxide. In addition, nano-TiO_2_ has great application potential in biomedical fields [3,4] due to the fact of its good antibacterial activity, favorable biocompatibility, and unique photocatalytic activity [5].

Nano-TiO_2_ nanostructures include titanium dioxide nanoparticles (TiO_2_-NPs) and titanium dioxide nanotubes (TNTs). In nature, TiO_2_-NPs mainly exist in the form of rutile, anatase, and brookite. Rutile is a stable phase, whereas anatase and brookite are metastable phases [6]. Anatase has the highest photocatalytic activity [7,8,9]. TNTs are one-dimensional hollow structures. Preparation methods mainly include template synthesis [10], anodic oxidation [11], and hydrothermal synthesis [12]. Two different TiO_2_ structures can produce reactive oxygen species to induce oxidative stress and destroy the cell walls of bacteria, thus exerting antibacterial activity and having strong mechanical properties in stomatology [13,14,15,16]. The most noteworthy are TNTs, which are considered to be ideal candidate materials for promoting the clinical therapeutic effects of medical implants among the various nanomorphological modifications of oral titanium (Ti) implants due to the fact of their enhanced biological activity and ability to achieve local drug elution [17,18].

The oral cavity’s physiological function and pathological changes are closely related to the health of other parts of the body. Therefore, higher safety requirements should be put forward for materials used in the oral cavity. Before any material can be used in the mouth, its biosafety and stability in human tissue must be fully understood. Currently, the biological toxicity of nano-TiO_2_ is considered to be related to its primary particle size, shape, agglomeration size, and other factors [19]. For example, the smaller the size of NPs, the more toxic they are thought to be [20,21]; needle- and short rod-shaped particles induce more cell damage than spherical- and long rod-shaped particles [22]. However, the existing experimental results and evidence do not specifically prove that nano-TiO_2_ has serious effects on human tissue. Nano-TiO_2_ has many advantages, acting as an oral cavity biomedical material and having huge application potential in stomatology (shown in Figure 1). Therefore, the application of nano-TiO_2_ in stomatology deserves more in-depth research.

In this review, we describe the characteristics and research status of applications of nano-TiO_2_ according to the different research fields of oral disease treatment, we evaluate the toxicological effects of its applications, and we analyze the prospects and challenges of its applications in stomatology. Our study should provide a basis for wider and safer applications of nano-TiO_2_ in the field of stomatology in the future.

## 2. Surface Modification of Dental Implants

Ti and its alloys have good corrosion resistance, mechanical strength, and biocompatibility, making them ideal materials for dental implants [23,24,25,26,27,28,29,30,31,32,33]. However, the lack of biological activity on the surface of natural Ti implants makes them highly prone to bacterial infections [32,34,35] and often causes insufficient bone tissue integration [14]. These issues limit the application of Ti in dental implants. A bacterial infection usually occurs within the first two weeks after implantation [36]. Bacteria adhere to and grow on the implant surface to form a biofilm, which hinders the role of the immune system [37] and is resistant to antibiotics. In addition, the physical, chemical, and biological properties of the implant surface affect the proliferation, adhesion, growth, and differentiation of cells which, in turn, affect the osseous integration of the implant with surrounding tissues [38,39,40,41,42]. Therefore, the surfaces of implants should be modified appropriately to enhance antibacterial activity, to inhibit the formation of biofilms, and to avoid the occurrence of peri-implant infections [25,43], while at the same time guiding the biological behavior of cells, improving bone integration, and improving the success rate of implant surgery.

Nano-TiO_2_ is one of the most studied metal oxides with antibacterial activity. It exhibits good bactericidal action against various Gram-positive and Gram-negative bacteria and fungi (e.g., *Escherichia coli* [15,16], *Staphylococcus aureus* [14,44,45], *Streptococcus mutans* [46], *Streptococcus sanguis* [47], and *Candida albicans* [48,49]) and, therefore, has potential for treating various oral infectious diseases [45,50] such as dental caries, periodontitis, dental pulp infection, and peri-implant inflammation [51]. Furthermore, TNTs can mimic the nanomorphology of the outer cell membranes of osteoblasts around implants, increasing the interaction between implant surfaces and neighboring cells, thereby enhancing osseointegration between native tissues and implant interfaces [52]. Therefore, nano-TiO_2_ is an ideal structure for implant surface coating [53]. At present, methods for preparing nano-TiO_2_ structural coatings on the surface of Ti and its alloy implants include an anodization technique [47,48], micro-arc oxidation [54], the sol-gel process [16], vapor deposition [44,55], pulse laser deposition [56], atomic layer deposition [37], and other methods.

On the surface of implants, nano-TiO_2_ exhibits a good bacterial killing effect due to the fact of its small size and strong oxidation capacity; it can be combined with other antibacterial metals (such as silver [16,57]) [46,51] to achieve synergistic antibacterial effects [14,57], and it can be combined with antibiotics to combat drug-resistant strains [58]. Furthermore, Zhang et al. prepared neutrophils containing photocatalytic TiO_2_-NPs in vivo, which fully mobilized the host’s defense mechanism and achieved an effective and powerful therapeutic response to pathogenic bacterial infection with low drug resistance and low virulence [59]. In addition to the enhanced antibacterial effect, an implant surfaces modified by nano-TiO_2_ also promoted the adhesion, proliferation, and growth of various mesenchymal stem cells (MSCs) [54,60], improving biocompatibility and bone integration [54,61].

To date, there have been several beneficial research results in the application of nano-TiO_2_ to implant surface modification. For example, after preparing TNTs on the surface of Ti implants by an anodization technique, Huang et al. obtained implant surfaces with enhanced hydrophilicity and MSC differentiation and a higher percentage of bone–implant contact (BIC), which showed great potential for clinical applications [62]. Baoe et al. first loaded TNTs with simvastatin (Sim), a drug that can promote bone formation, and then coated the nanotubes (NTs) with a thermosensitive chitosan/glycerin/hydroxypropyl methylcellulose hydrogel (CGHH) coating to control the release of simvastatin. As compared with the Sim@NT and NT groups, the Sim@CGHH group showed higher alkaline phosphatase (ALP) activity, which was conducive to the osteogenic differentiation of MC3T3-E1 cells, and the number of E. coli colonies was also lower (as shown in Figure 2) [63]. Yong et al. prepared a FAgHA-TiO_2_ (FagHA/TNT) nanocomposite double-layer coating on the surfaces of implants. The coating could simultaneously provide the advantages of both TiO_2_ and FagHA, and it had excellent antibacterial performance and cellular compatibility. Moreover, the anchoring effect of TNTs also increased the bonding strength of the coating by >17 MPa_2_ and the corrosion resistance by nearly two orders of magnitude [42]. Although nano-TiO_2_ in Ti implant surfaces has not been clinically applied due to the weak mechanical strength between it and Ti implants [64], in vitro studies have shown that nano-TiO_2_ can provide good surface topography to improve the clinical performance of dental implants. In the future, nano-TiO_2_ is expected to provide a promising surface modification strategy for improving the antimicrobial activity and biocompatibility of Ti implants with bone tissue.

## 3. Applications in Tissue Engineering and Regenerative Medicine (TERM)

Recently, the field of tissue engineering and regenerative medicine in dentistry has shown great potential in the treatment of craniofacial and tooth defects caused by trauma, tumor, or other diseases. The field aims to research and develop biosubstitutes to repair damaged tissue structures and functions using elements such as biocompatible scaffolds, stem cells, and growth factors [65]. Scaffolds are an important part of research in this area, because they can provide the optimal aperture range for the specific cells that stem cells produce, mimic the extracellular matrix, and provide the appropriate culture medium for cell growth. Studies have shown that the mechanical properties and biological activity of commonly used bone tissue engineering scaffold materializers (bioceramics, polymers, etc.) can be improved by adding nano-TiO_2_ [66,67] and can promote an increase in the production of mineralized matrix, making scaffolds with better biocompatibility and biological activity [67,68,69].

Although there have been many breakthroughs in the application of TERM in the oral cavity, it is difficult to achieve satisfactory bone integration after implantation due to the inherent biological inertia, stress shielding effects, and limited space for bone inward growth of Ti implants commonly used in clinics today [70,71]. Therefore, promoting the regeneration and integration of bone defects around oral implants remains an urgent problem to be solved. In view of this situation, existing implants should be properly modified to promote the development of regenerative medicine in the oral cavity and better benefit patients with oral diseases.

Nanomaterials play a significant role in craniofacial and dental tissue engineering. Among them, TNTs exhibit excellent biological activity, which can improve the biological behavior of osteoblasts [72,73], human periodontal ligament stem cells (PDLSCs) [69], human bone-marrow-derived mesenchymal stem cells (BMSCs) [51,54], and adipose-derived stem cells (ADSCs) [60,74], thereby promoting bone integration directly. In addition, they can facilitate the adhesion and proliferation of fibroblasts [72,75], human gingival epithelial cells (HGECs), and human gingival fibroblasts (HGFs) [76], making the soft tissue around an implant form a protective tissue barrier for potential bone integration. Therefore, nano-TiO_2_ can be directly incorporated into tissue engineering scaffolds to improve the mechanical properties of scaffolds and can also be applied to Ti implant coatings to provide effective surface modification [62,77]. For example, Roberta et al. prepared TNTs on implant surfaces and further modified them with polyelectrolyte multilayers (PEMs) based on Tanfloc (a cationic tannin derivative) and glycosaminoglycans (heparin and hyaluronic acid), increasing the rate of osteogenic differentiation and bone mineral deposition of ADSCs [78].

The osteogenic potential of TNTs is also reflected in their antioxidant properties [79]. Human osteogenesis is inhibited under oxidative stress [80], while nanotubes can effectively attenuate the negative effects of oxidative damage on osteogenesis via the synergistic effect of ITG α5β1 and the activation of Wnt signaling [81]. The size of TNTs affects the biological behavior of stem cells. Shen and Seunghan’s study showed that large TNTs were more conducive to the proliferation and differentiation of osteoblasts [82,83]. In addition, Yu’s study showed that small TNTs were beneficial to the adhesion and proliferation of osteoblasts in a normal microenvironment, while large TNTs increased osteogenic differentiation. After H_2_O_2_ treatment (simulating oxidative stress), only large TNTs showed the cellular behavior of increasing osteoblast adhesion, survival, and differentiation (as shown in Figure 3) [81], indicating that large TNTs were more suitable for preventing oxidative damage [84]. These finding have implications for bone integration on implant surfaces in people with systemic diseases (diabetes, osteoporosis, etc.).

## 4. Carrier for Drug Delivery

Targeted drug delivery and local drug delivery are considered to be the most forward-looking strategies to address the inherent limitations of traditional drug delivery [85]. The characteristics of oral diseases determine that treatment of them often requires local administration. The ideal oral local administration should provide sustainable and stable drug release, have a long-term therapeutic effect, and reduce the toxic side effects of drugs and medication frequency. Recently, advanced nanotechnology has produced various nanomaterials that are effective carriers of drugs and that are conducive to the efficient loading, targeted delivery, and controlled release of drugs. TNTs have become an ideal substrate for drug delivery in stomatology [86,87] due to the fact of their higher drug loading capacity and slower drug release kinetics [88] as well as their excellent chemical inertia, mechanical robustness, and good biocompatibility.

Bacterial infection is the main reason for the failure of implant surgery. In order to prevent an infection around an implant after surgery, a drug sustained-release delivery system that can provide continuous release of antibacterial drugs at therapeutic concentrations over 4–6 weeks should be mounted on an implant surface [89]. Numerous studies have shown that TNT modification and antibiotic loading can significantly enhance the antibacterial ability and osteogenic activity of implants [90,91]. However, as a drug delivery system, TNTs have the disadvantage of uncontrollable drug release [92]. Researchers have found that covering the surfaces of the TNTs with a polymer layer is a promising way to solve the problem of their sudden release. Chitosan (CS) is a biopolymer with wide application potential. Coating the CS layer on porous TNT arrays can effectively control the release rate of drugs by controlling the thickness and degradation kinetics of the CS film [93,94,95]. Seyed et al. first prepared a completely regular titania nanotube (cRTNT) array on a titanium substrate, then prepared chitosan nanofiber (CH) and reduced graphene oxide (RGO) double-layer coatings on the nanotube, and finally loaded vancomycin (VM) into the system for experiments. The results showed that the system could improve the drug burst release and prolong the release time, as well as improve the osteogenic and antibacterial activity (as shown in Figure 4) [94]. This drug delivery system, which uses TNTs as carriers to prepare multifunctional surfaces through reasonable assembly of components with certain characteristics, has been used for loading and releasing a variety of antibiotics [17,92,96], indicating a promising direction for the development of advanced drug delivery systems.

In addition to CS, TNTs modified in other ways can also play an important role in drug delivery [97]. Baoe et al. found that the incorporation of AgNPs into TNTs showed valuable biological and time-dependent antibacterial properties. In the early stage, TNTs exhibited strong “release sterilization” activity that could prevent an initial infection after surgery. Then, they intelligently changed to exhibit “contact sterilization”, thereby protecting implants from chronic infection, reducing the biosafety problems of AgNPs, and meeting various antibacterial requirements in different periods after biomaterial implantation [36]. Dong et al. prepared a pH-dependent AgNP-releasing implant through transplanting AgNPs onto the surface of an implant modified with TNTs via a low pH-sensitive acetal joint (TNT-Al-AgNPs). In the case of bacterial infection, the pH of the surface around the implant was reduced from 7.4 to 5.5 due to the bacterial metabolism and acid production, inducing the implant to release a higher dose of AgNPs than under physiological conditions, which increased the antibacterial efficiency of *Staphylococcus aureus* and *Enterobacter coli* by 12.7 times and 5.1 times, respectively, compared to that without infection, and it also enhanced the proliferation and differentiation of osteoblasts [98]. The photocatalytic activity of Nano-TiO_2_ can only be triggered under ultraviolet (UV) irradiation, but UV has a limited penetration depth in tissue and can cause photodamage to biological tissues. Zhao designed a near-infrared (NIR) controlled drug delivery system with two hydrophilic structures using upconversion (UC) correlation strategies. The system triggered the photocatalytic activity of TiO_2_-NTs through NIR and realized the controllable release of drugs; therefore, the hydrophobic monolayer on the surfaces of NTs could effectively reduce the toxicity of reactive oxygen species (ROS) on healthy skin cells, broadening the biological application of nano-TiO_2_ [99]. These results indicate that TNTs can be used as a promising material for an oral medicine drug delivery system.

## 5. Additives in Dental Materials

TiO_2_-NPs are ideal additives for enhancing the properties of polymer materials owing to their unique photocatalytic activity and chemical stability. As promising additives for dental materials, TiO_2_-NPs mainly improve the antibacterial properties and mechanical strength of dental materials [100,101]. TiO_2_-NPs have broad-spectrum antibacterial activity against microorganisms, with a noncontact bactericidal role [102], and they can be used as antibacterial fillers for dental composites [103]. Kuroiwa et al. applied a nano-TiO_2_ coating on orthodontic resin to develop an orthodontic resin with antibacterial properties, and it achieved satisfactory results [104]. Moreover, the addition of TiO_2_-NPs has been shown to improve the vinyl conversion degree of a resin [105] and to remarkably upgrade mechanical properties such as bending strength and hardness [106], thus enhancing the bond strength of the binder to teeth [107].

There have been many beneficial explorations of TiO_2_-NPs as additives to enhance the antimicrobial properties and mechanical strength of dental materials. Two striking examples include poly(methyl methacrylate) (PMMA) and resin-modified glass ionomer cements. PMMA is one of the most widely used materials in the oral cavity, but its porous surface (conducive to microbial adhesion) and weak mechanical properties (leading to wear or fracture) are major problems in its application [108]. Adding TiO_2_-NPs to PMMA can improve its mechanical stiffness, wear resistance, and fracture resistance, and it can reduce its roughness. The *C. Albicans* yeast colonization percentage of PMMA with 1% and 3% TiO_2_-NPs decreased by 22% and 26%, respectively, after 48 h compared with PMMA without TiO_2_-NPs [109,110]. Fully edentulous patients with 3D-printed dentures showed significantly increased satisfaction in aesthetic, masticatory efficiency, and comfort, which maintained their improved characteristics after use for 18 months [111]. The weak mechanical strength and toughness of glass ionomer cement (GIC) are the main problems for permanent repair. The incorporation of TiO_2_-NPs into GIC increased the particle size distribution and occupied the blank area between GIC particles to inhibit the propagation of cracks, thus enhancing the strength of the material [112]. The mechanical properties [113] and antibacterial properties [114] of the material were upgraded without affecting the bonding with enamel and dentin [115,116].

## 6. Assistance in the Diagnosis and Treatment of Oral Tumors

Oral cancer is a common malignant tumor of the head and neck, which is ranked as the sixth most common cancer in the whole body. Thus, simple, rapid, and accurate diagnostic tools are important for clinical diagnosis and treatment of tumors. Raman spectroscopy has been successfully used to detect tumor diseases in different parts of the body [117]. Nano-TiO_2_ has attracted extensive attention in the development of surface-enhanced Raman scattering (SERS) substrates because of its easy growth and controllable nanostructure array [118]. Girish et al. constructed a catheter device with an SERS substrate consisting of foliated nano-TiO_2_ modified using AgNPs. The SERS, composed of closely stacked adjacent foliated TiO_2_ nanostructures and AgNPs, helped to form more “Raman” hot spots and could rapidly detect, classify, and grade normal, precancerous, and malignant tissues with high sensitivity and a high accuracy of 97.84%. The average detection time for each patient was only 25–30 min, which helped to improve the application effect of Raman spectroscopy in oral cancer detection [119].

At the early stage of malignant progression, circulating tumor cells (CTCs) can break away from original or metastatic tumors and then invade a distal site in different tissues of the body, which is the main route of cancer metastasis. Therefore, tumor progression can be determined by detecting CTCs, but CTCs are difficult to accurately detect and isolate as a result of their phenotypic heterogeneity and rarity [120]. Due to the fact of their large specific surface area, nanomaterials can enhance cell adhesion and, therefore, enhance the capture affinity and sensitivity of CTCs [121]. Nano-TiO_2_ has great potential in efficiently and sensitively capturing CTCs [122]. The capture and release efficiencies of CTCs using a platform made of nano-TiO_2_ were 92.9% and 89.9%, respectively, which was helpful for further diagnosis and treatment of tumors (the process of nano-TiO_2_ modification and the capture and release of CTCs are shown in Figure 5) [123]. These studies show the potential of nano-TiO_2_ applications in the diagnosis of oral tumors.

In addition to its diagnostic application for oral cancer, nano-TiO_2_ can cause cytotoxicity and oxidative stress of cancer cells and can stimulate the production of ROS for cell killing; therefore, it has good anticancer activity [52]. TiO_2_-NPs biologically modified by herbs show good anti-KB oral cancer cell performance and are also less toxic to normal cells [124,125]. In recent years, photodynamic therapy (PDT) based on photosensitizers that are activated to produce ROS after being irradiated with a specific wavelength of light to inhibit cancer cells has aroused great interest among scholars. However, due to the limited penetration depth of visible light, traditional PDT is limited to the treatment of superficial and flat lesions [126]. As a potential photosensitizer, TiO_2_-NPs exhibit excellent UV-light induced cytotoxicity [127]. Upconversion nanoparticles (UCNs) have been embedded into TiO_2_ matrix to improve the photocatalytic effect of TiO_2_, showing great potential for improving the penetration depth limit of conventional PDT and for expanding the application of PDT to thick and solid advanced or recurrent head and neck cancers [128]. Currently, the use of nano-TiO_2_ in the diagnosis and treatment of cancer has involved a number of cell and mouse experiments [129]; for example, a therapeutic diagnostic platform consisting of TiO_2_-NPs doubled the survival rate of mice with multiple myeloma (MM), a malignant plasma cell disease of bone marrow origin [130]. In the future, more attention should be given to in vivo and clinical trials, and targeted research on oral cancer should be carried out, striving for applications of this nanomaterial for oral cancer clinical treatment as soon as possible.

## 7. Prospective Applications and Challenges of Nano-TiO_2_ in Dentistry

Nano-TiO_2_ has stable physicochemical properties, is inexpensive and easy to obtain, and has good biocompatibility; therefore, it is a research material that is considered to be significant in stomatology. The excellent antibacterial activity and biological activity of nano-TiO_2_ provide a novel method for implant surface modification and tissue engineering. Its higher drug loading capacity and slower drug release kinetics make it a good carrier for oral drug delivery. Its strong antibacterial and mechanical properties make it a useful additive for dental materials. Moreover, its larger specific surface area can assist in the diagnosis of oral diseases.

Nevertheless, most studies on nano-TiO_2_ are currently performed in vitro, and more information regarding the clinical outcomes of toxicity and biocompatibility is needed for careful evaluation before it is applied to clinical practice. At present, humans are mainly exposed to nano-TiO_2_ through oral, inhalation, and skin contact; the oral route is the main type of exposure. In mouse experiments, after intragastric administration, TiO_2_-NPs were absorbed by the gastrointestinal tract [131] and accumulated in the spleen and liver [132,133]. TiO_2_-NPs have been shown to damage multiple organs of mice (intestine [134], liver [135], spleen [136], kidney [137], etc.) by inducing cell injury and changing the expression of inflammatory cytokines [138,139,140,141]. In addition, TiO_2_-NPs have been reported to penetrate the placental barrier to induce developmental toxicity [142] and the blood–brain barrier (BBB) to induce neurotoxicity [22]. NPs deposited in the brain may induce oxidative stress imbalance, resulting in DNA damage and neurodegeneration, causing mice to exhibit significant behavioral deficits [143]. Intranasally administered TiO_2_-NPs have been reported to accumulate in multiple organs (i.e., liver, spleen, kidney, brain, stomach, and heart) via pulmonary transport. High doses of TiO_2_-NPs have been shown to cause or exacerbate some respiratory diseases [144,145], whereas long-term and low-concentration exposure (continuous exposure of A549 alveolar epithelial cells to 1–50 μg/mL TiO_2_-NPs over 2 months) to TiO_2_-NPs did not affect the cell viability of A549, but accumulation of TiO_2_-NPs in the cells resulted in DNA damage, reduced cell proliferation rates, and caused an allergic response to methane methylsulfonate (MMS) [146]. After skin exposure, TiO_2_-NPs were detectable in the stratum corneum layer of the epidermis and follicular epithelium but neither in the viable skin tissue nor in the internal organs (i.e., brain, liver, spleen, and kidney) [147]. There is no evidence of carcinogenicity, mutagenicity, or reproductive toxicity after skin exposure to nano-TiO_2_ [148].

Due to the lack of reliable biosafety models, further studies on the biosafety of nano-TiO_2_ are needed in the future. How to correctly and rationally use nano-TiO_2_ is a challenge for researchers. Nevertheless, if we fully consider and prudently use nano-TiO_2_ in the treatment of oral diseases, we believe it could significantly improve the therapeutic effect.

## 8. Conclusions

Nano-TiO_2_ has low production costs, good physicochemical properties, and stable properties. Due to the fact of its photocatalytic sterilization and biocompatibility, it has great potential for the treatment of oral diseases. However, research on its application in oral disease treatment is still at the stage of cell and bacterial experiments in vitro and animal experiments in vivo, and currently there are no convincing clinical experimental results. There is still a long way to go before this nanomaterial can be applied in real-time clinical practice, and more investigative experiments are needed. In the future, while continuing to explore potential applications of nano-TiO_2_ in stomatology, researchers also need to further explore methods to reduce its toxicity and to improve its mechanical stability and antibacterial effect. In addition, appropriate biological models need to be established as soon as possible for clinical research on the use of nano-TiO_2_ to improve the oral health of the population.

## Figures and Tables

**Figure 1 molecules-27-03881-f001:**
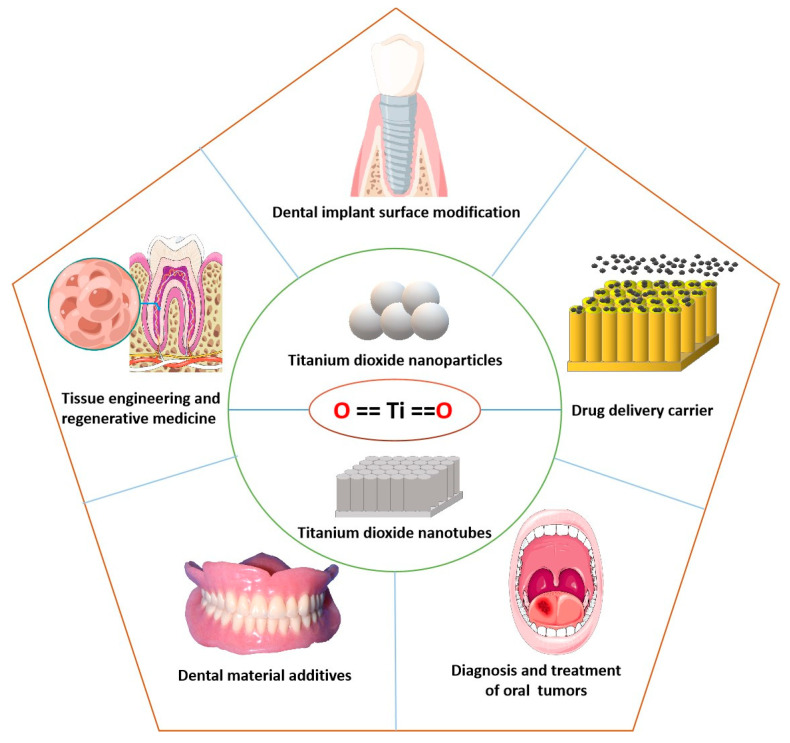
Common forms of nano-TiO_2_ and its application in stomatology (the figure was drawn using Figdraw, and part of picture material is cited from https://smart.servier.com/ (accessed on 13 May 2022)).

**Figure 2 molecules-27-03881-f002:**
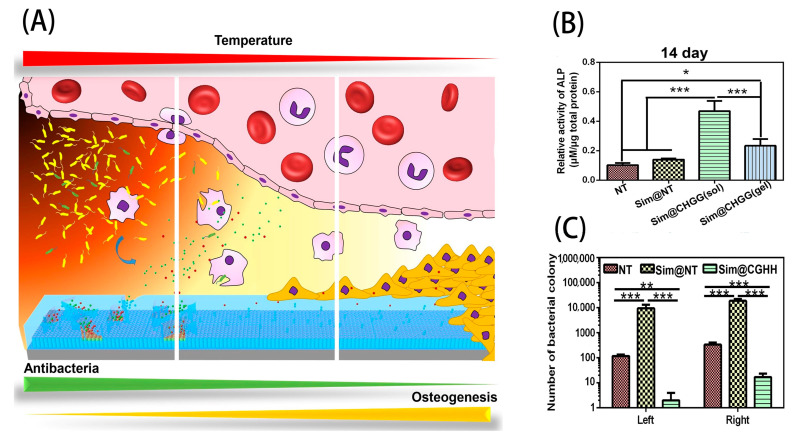
(**A**) The constructed coating. At a normal body temperature (37 °C), the hydrogel is in a sol state, which controls the continuous release of simvastatin and promotes long-term osteogenic differentiation. When the temperature rises to 40 °C, the hydrogel changes from sol to gel, releasing Gly to stimulate macrophages to polarize into a proinflammatory M1 phenotype to kill bacteria. (**B**) The results of the ALP activity of MC3T3-E1 after 14 days of culture. (**C**) *E. coli* colony count. Reprinted from Reference [63]; Copyright 2022, with permission from Elsevier. * *p* < 0.05; ** *p* < 0.01; *** *p* < 0.001.

**Figure 3 molecules-27-03881-f003:**
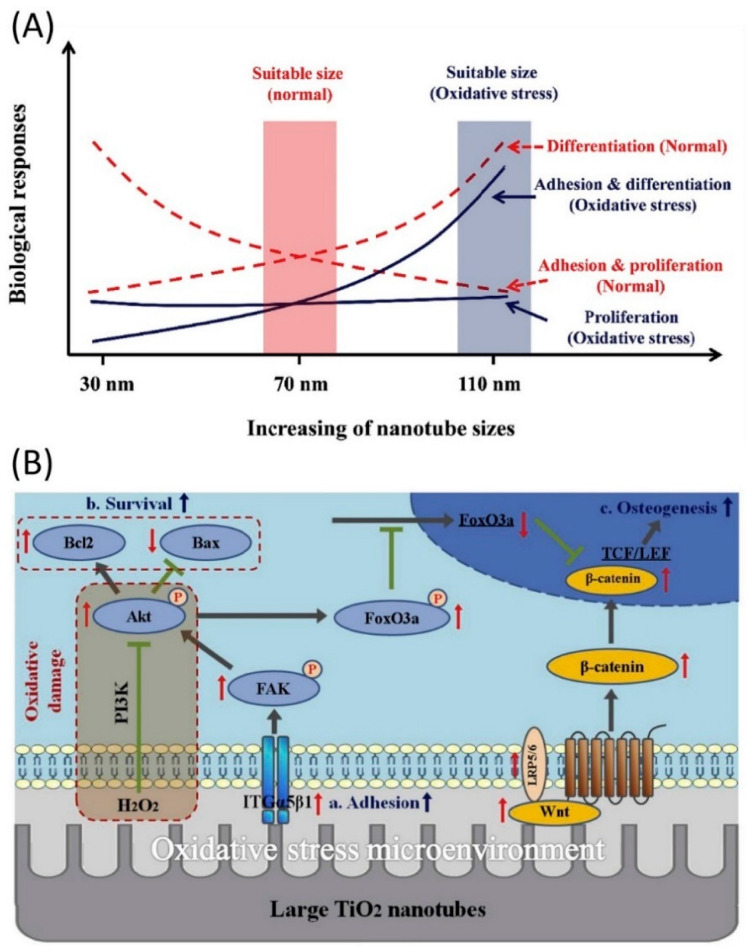
(**A**) Cell behavior in response to TNTs of different sizes in different microenvironments; (**B**) protective effect of large TNTs on ROS injury. Reprinted from Reference [81]; Copyright 2022, with permission from Elsevier. (a. The high expression of ITG α5β1 on TNT_110_ substrates promoted the early adhesion of osteo-blasts; b. The up-regulation of Bcl2 and down-regulation of Bax improved cell survival; c. High expression of p-FoxO3a and β-catenin proteins promoted the osteogenic differentiation).

**Figure 4 molecules-27-03881-f004:**
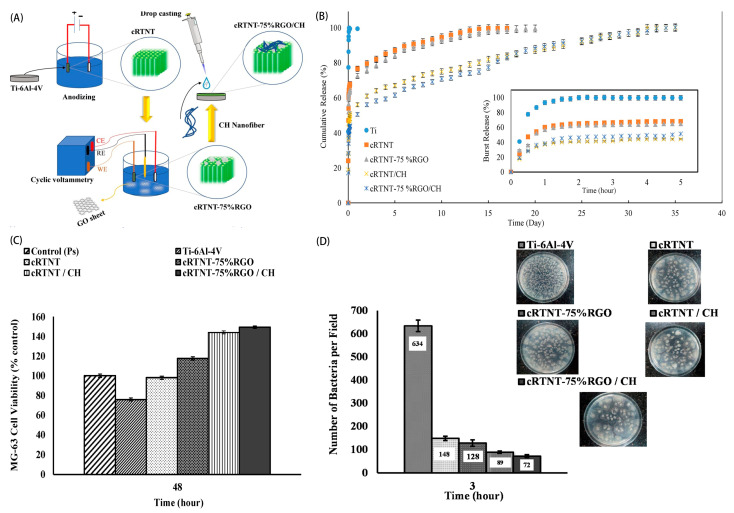
(**A**) Construction diagram of a nanofibrillated, chitosan-coated, highly ordered titania nanotube array/graphene nanocomposite; (**B**) release curves of vancomycin in different groups; (**C**) viability of MG63 cells in different groups; (**D**) colony count of Staphylococcus aureus in different groups. Reprinted from Reference [94]; Copyright 2022, with permission from Elsevier.

**Figure 5 molecules-27-03881-f005:**
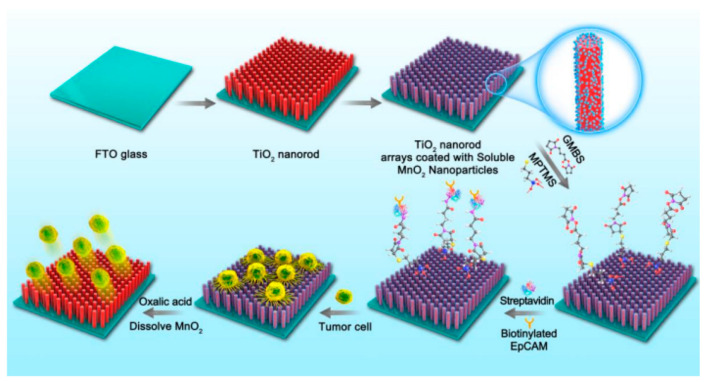
Modification of the substrate to capture and release CTCs. Reprinted with permission from Reference [123]; Copyright 2022, American Chemical Society.

## Data Availability

Not applicable.

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
