# Peer review of "Applications of Titanium Dioxide Nanostructure in Stomatology"

_molecules, 2022, doi:10.3390/molecules27123881_

Round 1

Reviewer 1 Report

I have reviewed the manuscript titled, "Application of titanium dioxide nanostructure in stomatology". It is my view that the content is adequate for further consideration and acceptance after language editing. I noticed several typos in the manuscript, so it is in the best interest of the authors to revise it for clarity. 

Reviewer 2 Report

The manuscript is addressing bio-safety of Titanium oxide nanomaterials  mainly targeted for dental applications (stomatology). The motivation of the review seems to be appropriate and should have significant potential readership.

However, I do not recommend the paper for publication in Molecules based on the following reasons.

1. English writing should be extensively edited. Current version of the manuscript is not easy to understand.

2. Quality of figures are so low that the readability is significantly hampered.

Reviewer 3 Report

The article entitled “Application of titanium dioxide nanostructure in stomatology”, the aim of this review: the characteristics and research status of the application of Nano-TiO2 according to the different research fields of oral disease treatment were described, the effects of toxicity on its application were evaluated, and the prospect and challenges of its application in stomatology were analyzed.

Below are some suggestions:

In the Abstract:

- I suggest focusing more on writing the abstract according to the purpose of the review, also inserting an accurate conclusion in view of the observed data.

In the Introduction:

- The introduction is well written and objective, as in figure 1.

2. Surface modification of dental implants

- In last paragrafh: “These studies show that TNTs provide a good surface morphology for improving the clinical performance of dental implants, and a promising surface modification strategy for clinical applications”........... Detail what these clinical applications would be, demonstrating the clinical results obtained?

- I suggest improving the quality of figure 2

3. Applications in tissue engineering and regenerative medicine (TERM)

- In the first paragrafh: “Scaffolds are an important part of research in this área”....... In fact, within TERM there are many studies involving scaffolds, however, in this case, would they be associating TNTs with any specific one? Or only on implant surfaces as described in the previous item?

4. Carrier for drug delivery

- I suggest separating figure 4 into: A and B / C and D to improve quality and visualization.

6. Assistance in the diagnosis and treatment of oral tumors

- I would like more comments and explanations about the clinical applicability.

8. Conclusions

- The conclusion could be in accordance with the objective of the research, I believe it was too superficial and I suggest emphasizing the clinical part.

Bel

Round 2

Reviewer 2 Report

Both of figure quality and English writings are significantly improved in the current version of manuscript. However, it seems like that basic level of spelling check process has not been performed as can be seen in some sentences, for example, 'nanostructuresinclude' in the first page, and some of TiO2 instead of TiO2.

I recommend the authors to print out their manuscript and read it thoroughly for the final check to shape it out for publication, which should be duty for the authors. 

I recommend the manuscript to be published in Molecules as a review article after minor revision for spell checks.

Author Response

Dear Reviewer,

Thank you very much for your careful reading and helpful suggestions, and
we are very sorry for the mistakes in this manuscript. We have checked the
manuscript carefully and revision the mistakes. We hope the manuscript can
meet the journal's standard.

Best regards